# Structural and Textural Ultrasound Features of Gastrocnemius Medialis in Chronic Stroke: Associations with Functional Outcomes and Spasticity

**DOI:** 10.3390/jcm14217680

**Published:** 2025-10-29

**Authors:** Clara Pujol-Fuentes, Juan Nicolas Cuenca-Zaldívar, Mª Dolores Navarro Pérez, Kristin Musselman, Francisco Álvarez-Salvago, Pablo Herrero, Samuel Fernández-Carnero

**Affiliations:** 1FIBIO Research Group, Department of Physiotherapy, Faculty of Health and Sciences, Universidad Europea de Valencia, 46010 Valencia, Spain; clara.pujol@universidadeuropea.es (C.P.-F.); salvagofran@gmail.com (F.Á.-S.); 2iHealthy Research Group, Instituto de Investigación Sanitaria (IIS) Aragon, University of Zaragoza, 50009 Zaragoza, Spain; 3Grupo de Investigación en Fisioterapia y Dolor, Departamento de Fisioterapia, Facultad de Enfermería y Fisioterapia, Universidad de Alcalá, 28801 Alcalá de Henares, Spain; j.nicolas7251@gmail.com (J.N.C.-Z.); samuel.fernandezc@uah.es (S.F.-C.); 4Research Group in Nursing and Health Care, Puerta de Hierro Health Research Institute—Segovia de Arana (IDIPHISA), 28222 Majadahonda, Spain; 5IRENEA—Instituto de Rehabilitación Neurológica, Fundación Hospitales Vithas, Callosa d’En Sarrià, 46007 Valencia, Spain; lolesnavarro@gmail.com; 6Department of Physic al Therapy and Rehabilitation Sciences Institute, Temerty Faculty of Medicine, University of Toronto, Toronto, ON M5S 1A8, Canada; kristin.musselman@utoronto.ca; 7KITE Research Institute, Toronto Rehabilitation Institute, University Health Network, Toronto, ON M5G 2A2, Canada; 8Department of Physiatry and Nursing, Faculty of Health Sciences, University of Zaragoza, 50009 Zaragoza, Spain

**Keywords:** stroke, ultrasonography, muscle architecture, echotexture, function, spasticity, gastrocnemius medialis

## Abstract

**Background/Objectives**: Stroke is a leading cause of disability, and post-stroke spasticity frequently impairs ankle mobility, strength, and gait. The gastrocnemius medialis (GM) is central to these deficits, yet the relationship between its ultrasound characteristics, functional outcomes, and spasticity severity remains unclear. This study aimed to compare structural and textural ultrasound features of the GM between individuals with chronic stroke presenting ankle spasticity and healthy controls, and to examine their associations with functional performance and spasticity severity. **Methods**: This case–control study included 26 individuals with stroke and 26 matched controls. Ultrasound assessments were performed using B-mode imaging to obtain parameters such as muscle thickness, pennation angle, and textural features (first-, second-, and higher-order). Functional measures included mobility (Timed Up and Go), walking speed (10-Meter Walk Test), ankle strength (dynamometry), and range of motion (goniometry). Spasticity was evaluated separately using the Modified Ashworth Scale. **Results**: No significant differences in GM ultrasound parameters were observed between groups or limbs (*p* > 0.05). Participants with stroke showed significantly reduced dorsiflexion mobility and lower strength for both plantarflexors and dorsalflexors. Correlations between ultrasound parameters and functional measures were not statistically significant; however, the effect size was consistently small. Spasticity severity did not significantly influence ultrasound findings. **Conclusions**: GM ultrasound parameters did not distinguish participants with stroke from controls or meaningfully correlate with function or spasticity. Functional impairments may stem primarily from neural mechanisms and compensatory motor strategies rather than muscle alterations detectable by ultrasound.

## 1. Introduction

Stroke remains one of the leading causes of mortality, morbidity, and long-term disability worldwide. Globally, the burden of stroke is substantial. According to the Global Burden of Disease Study, stroke ranks as the third leading cause of death and the fourth in terms of disability-adjusted life years (DALYs) lost, accounting for approximately 7.3 million deaths and over 160 million DALYs globally [1].

Stroke imposes a tremendous cost on healthcare systems, families, and societies. In 2017, the total economic burden of stroke in the European Union was estimated at €57 billion [2]. This includes direct healthcare costs, as well as indirect costs related to lost productivity and long-term disability. The presence of post-stroke spasticity significantly amplifies care costs, with studies indicating up to a fourfold increase compared to patients without spasticity [3]. These findings highlight the need to optimize stroke-related healthcare expenditures through effective prevention, early diagnosis, and rehabilitation strategies [4,5].

Among the sequelae of stroke, limited ankle range of motion [6], reduced strength [7], and spasticity [8] are highly prevalent and contribute substantially to functional limitations. In the lower limb, especially around the ankle joint, spasticity often leads in deformities such as equinus, varus, or equinovarus foot postures [7,9], stemming from the combined effects of increased muscle tone and weakness of the surrounding musculature during gait [7]. These alterations impair gait [10], compromise balance and independence, and are associated with an increased risk of falls [11] and reduced quality of life [12]. The gastrocnemius medialis (GM) plays a critical role in these deformities due to its insertion and function. Its contribution to plantarflexion and foot inversion links it to the development of equinovarus patterns, making it a prime candidate for focused evaluation in post-stroke rehabilitation.

Muscle ultrasound has emerged as a promising technique for assessing structural and compositional changes in neuromuscular disorders [13,14,15]. Compared to Magnetic Resonance Imaging (MRI) and Computed Tomography (CT), ultrasound offers real-time imaging, portability, and absence of ionizing radiation [16,17,18,19]. Importantly, ultrasound can detect architectural changes [13,20] (e.g., muscle thickness, pennation angle) and echotextural features [21,22] (e.g., echogenicity, echointensity) in muscles with neurological impairment, with some emerging methods such as histogram-based patch analysis also being explored [23]. Additionally, machine learning approaches are increasingly being applied to ultrasound imaging, enabling automated feature extraction and pattern recognition [24]. However, the integration of advanced texture analysis techniques—such as Gray Level Co-occurrence Matrix (GLCM), Grey Level Run-Length Matrix (GLRLM), and Grey Level Size-Zone Matrix (GLSZM)—into the evaluation of spastic muscles remains underexplored.

Some recent studies have explored the relationship between ultrasound-derived parameters and function [25], muscle strength [25,26,27,28] and spasticity [25]. In stroke populations, in general, muscle thickness and cross-sectional area have been associated with measures of functional performance and strength, although findings vary depending on the muscle assessed and the stage of the post-stroke recovery [25,26]. For spastic muscles, muscle thickness and echogenicity have shown no correlation with spasticity severity [25] and no studies to date have examined these variables stratified by different spasticity grades.

The main objective of this study is to determine whether structural and textural ultrasound features of the gastrocnemius medialis differ between the affected and unaffected limbs in chronic stroke patients, as well as between the affected limb and the dominant and non-dominant limbs of age and sex matched healthy controls. Secondary objectives were to examine the associations of ultrasound parameters with functional outcomes—including mobility, walking speed, ankle strength, and range of motion—as well as spasticity, and to evaluate differences in ultrasound and functional measures according to spasticity severity.

## 2. Materials and Methods


**Study Design**


This cross-sectional, observational and comparative study with correlational analyses was conducted between May and December 2024. The study followed the Strengthening the Reporting of Observational Studies in Epidemiology (STROBE) guidelines for observational studies [29].

A total of 75 individuals were initially contacted, from which 26 individuals with chronic stroke and 26 demographically matched controls (by biological sex and age ±3 years) were ultimately included. The methodology followed recommendations from previous studies [30].

Patients with a stroke diagnosis were recruited at Hospital Clínico Universitario Lozano Blesa in Zaragoza by a specialist in physical medicine and rehabilitation. Additional patients were recruited at Hospital Vithas in Valencia by a neurologist, through the Instituto de Rehabilitación Neurológica (IRENEA). Eligible patients were informed about the study objectives, procedures, and potential risks and benefits. Those interested in participating received an informed consent form and had the opportunity to ask questions before scheduling an assessment appointment at either the Lozano Blesa Hospital (Instituto de Investigación Sanitaria de Aragón) or Instituto de Rehabilitación Neurológica (IRENEA), depending on the referring center.

Control participants, with no history of stroke, were recruited from the researcher’s close environment. Upon verbal agreement, they were invited to sign the informed consent form and proceed with the evaluation. All participants provided written informed consent.

A physiotherapist with over 12 years of clinical experience conducted all assessments, lasting approximately 75 min. Each participant underwent a one-time evaluation that included anthropometric, sociodemographic, and clinical data collection, ultrasound image acquisition and assessments of functional mobility, walking speed, ankle strength, range of motion, and spasticity.


**Sample Size**


Sample size was calculated based on distal medial gastrocnemius thickness data reported by Picelli et al. [31], using a Student’s t-test for independent samples. Assuming α < 0.05, a statistical power of 80%, and an effect size of Cohen’s d = 0.86, a minimum of 52 participants (26 per group) was required. The larger of two estimates (affected vs. unaffected leg within participants with stroke and affected leg vs. control leg) was used to ensure robust statistical validity.


**Ethics Statement**


The study was approved by the Ethics Committee of the Autonomous Community of Aragón (CEICA) under code C.P.—C.I. PI24/030 and was registered on ClinicalTrials.gov (NCT06411587, https://clinicaltrials.gov/study/NCT06411587 accessed on 28 October 2025). All procedures complied with the Declaration of Helsinki 2024 [32].


**Participants**


The inclusion criteria of the stroke group were: (1) age ≥ 18 years; (2) diagnosed with a first unilateral ischemic or hemorrhagic stroke (excluding subarachnoid hemorrhage) confirmed via CT or MRI; (3) at least 6 months post-stroke; (4) plantarflexor spasticity graded 1, 1+, 2, or 3 on the Modified Ashworth Scale; (5) independent ambulation (with or without assistance); (6) ability to understand Spanish language. The exclusion criteria were: (1) other neurological conditions; (2) musculoskeletal disorders affecting the lower limb; (3) lower-limb surgery; (4) cognitive impairment; and (5) medical conditions interfering with study procedures.

The inclusion criteria for the control group were: (1) age ≥ 18 years; (2) independent ambulation (with or without assistance); (3) comprehension of Spanish language; (4) sex and age matched (±3 years) to stroke participants. The exclusion criteria: (1) History of stroke or spasticity; (2) neurological or musculoskeletal disorders; (3) lower-limb surgery; (4) cognitive impairment; (5) other interfering medical conditions.

In this study, the term “healthy controls” refers to the absence of stroke, acknowledging that this does not imply the complete absence of other medical conditions.


**Variables**


The variables analyzed were divided into descriptive variables (anthropometric, sociodemographic and clinical characteristics) and variables of interest, including one primary variable and several secondary variables. Anthropometric data comprised age, weight, height, and dominant side, while demographic data included biological sex, educational level, marital status, and whether the participant lived alone. Clinical characteristics included tobacco use and the number of cigarettes per day, alcohol consumption, current exercise habits, treatment services received, Visual Analogue Scale (VAS-10) scores for anxiety and depression, and physical activity (International Physical Activity Questionnaire (IPAQ) [33]. All participants completed an initial questionnaire administered by a physiotherapist at the start of the study. For individuals with stroke, additional clinical information was recorded: the hemiparetic side, use of assistive devices, history of botulinum toxin or dry needling in the gastrocnemius, and spasticity medication. Patients’ perception of ankle spasticity interference in daily life, difficulties with self-care, limitations in mobility and social participation, as well as previous exercise habits before stroke, were all assessed using yes/no questions. Finally, the level of spasticity was evaluated using the Modified Ashworth Scale (MAS) [34].

The primary variable of interest was the medial gastrocnemius muscle thickness (structural ultrasound parameter), measured via B-mode ultrasound using a portable device Butterfly iQ+ (Butterfly Network, Inc.), Burlington, Massachusetts, United States. Ultrasound imaging was performed with 50% gain and 5 cm depth. Participants were seated with knees at 90° flexion. The probe was placed at 30% of the distance between the medial tibial condyle and the medial malleolus, at a 35–45° angle relative to the horizontal plane [35]. Three transverse and three longitudinal images were acquired per participant.

The secondary variables included another structural ultrasound parameter, pennation angle, and a set of textural ultrasound features extracted from manually selected regions of interest (ROIs), after denoising with a median low-pass filter, performed using a macro applied to all image stacks in Fiji software (version 2.3.0, National Institutes of Health, Bethesda, MD, USA) [36]. These textural features comprised first-order (e.g., echogenicity, echovariation, entropy, skewness), second-order (derived from Grey-Level Co-occurrence Matrices (GLCM), Grey-Level Run-Length Matrices (GLRLM), and Grey-Level Size Zone Matrices (GLSZM), and higher-order features (Local Binary Patterns (LBP) and blob analysis). All features were calculated using the R packages *radiomic* [37,38], *EBImage* [39], and *wvtool* [40].

Additional secondary variables of interest included the level of plantarflexor spasticity, assessed using the MAS [34]; passive ankle dorsiflexion and plantarflexion range of motion, measured by goniometry [41]; ankle muscle strength during maximal isometric contractions of dorsiflexion and plantarflexion, assessed with a MicroFET2 dynamometer [41,42]; and functional mobility and walking speed, evaluated using the Timed Up and Go (TUG) [43] and the 10-Meter Walk Test (10MWT) [44].

For the MAS assessment of ankle plantarflexors, participants were placed in a supine position, and the examiner manually moved the ankle into dorsiflexion rapidly when possible, following the procedure described by Bohannon and Smith (1987) [45]. The examiner’s hand was placed on the calcaneus and the forearm along the plantar surface of the foot to control the movement and detect resistance during passive stretch.

Passive range of motion of ankle dorsiflexion and plantarflexion was measured in the supine position, with participants barefoot and starting from 0° of ankle dorsiflexion, positioning the tibia perpendicular (90°) to the plantar surface of the foot.

For dynamometry, plantarflexion was measured in a seated position [46] to avoid the influence of foot weight. A 20° knee flexion was added using a wedge to reduce isometric coactivation of the quadriceps and hamstrings that could interfere with the measurement. The dynamometer was placed against the wall and aligned with the metatarsal heads. Dorsiflexion strength was assessed in the supine position [46], maintaining the same 20° knee flexion setup.

The TUG [43] was performed by asking participants to stand up from a chair, walk 3 m, turn around, return, and sit down, recording the total time to complete the sequence. The 10MWT [44] was carried out as described in the standard protocol, allowing 2 m for acceleration and 2 m for deceleration, timing only the central 8 m.


**Blinding Procedure**


While full blinding was not possible during assessment due to the visible presence of spasticity, image analysis was performed by a blinded evaluator. All data were pseudonymized using randomly generated codes with identifiers for image orientation. To determine intra-rater reliability, repeated measurements were performed by the same experienced physiotherapist within a single evaluation session.


**Statistical Analysis**


Statistical analyses were performed using R software (version 4.1.3; R Foundation for Statistical Computing, Vienna, Austria). Texture metrics were averaged across five angular directions (0°, 45°, 90°, 135°, and 180°).

Normality was assessed using the Shapiro–Wilk test. Continuous variables were expressed as mean ± standard deviation, and categorical variables as absolute and relative frequencies. Between-group differences were analyzed using the Mann–Whitney U test or Fisher’s exact test, as appropriate.

Intra-rater reliability was assessed using a two-way mixed-effects model, single measures, consistency definition [ICC(3, 1)], along with the standard error of measurement (SEM) and minimal detectable change (MDC). Robust linear models with bootstrap resampling were used to assess differences between limbs and groups, adjusting for relevant covariates. Covariates showing multicollinearity (r > 0.70) were excluded [40].

Pearson or polyserial correlations were used to examine associations between ultrasound parameters and functional outcomes. Correlation strength was classified as negligible (<0.29), low (0.30–0.49), moderate (0.50–0.69), high (0.70–0.89), or very high (>0.90) [47].


**Data Protection**


Data handling adhered to General Data Protection Regulation (2016/679) and Spanish Organic Law 3/2018 on Personal Data Protection and Digital Rights. All procedures concerning data collection, storage, and confidentiality were described in the participant informed consent form. Pseudonymization was performed by a non-study iHealthy team member, who assigned randomly generated numeric codes to each participant before data analysis. Only the principal investigators had access to the linkage file. Research data will be securely stored for five years after publication, in accordance with institutional and legal requirements, ensuring full compliance with data protection regulations.


**Funding and Personnel**


The study was unfunded. Equipment was provided by the iHealthy research group. Data collection was conducted by a physiotherapist, supervised by a neurologist and a physical medicine and rehabilitation physician.

## 3. Results

### 3.1. Anthropometric, Sociodemographic and Clinical Characteristics

The anthropometric, sociodemographic and clinical characteristics of the sample are shown in Table 1. Significant differences were found in educational level (*p* = 0.001), with participants with stroke presenting a higher proportion of individuals without university education compared to healthy controls. Regarding lifestyle habits, participants with stroke reported lower alcohol consumption after their stroke (*p* = 0.005). They also showed greater depressive symptoms as measured by the *VAS-10 scale* (*p* < 0.001) and lower physical activity level according to the IPAQ (*p* = 0.008).

For the remaining variables, no significant differences were observed between groups, indicating that participants with stroke and healthy controls can be considered comparable in these aspects.

The clinical profile of participants with stroke (n = 26) is summarized in Table 2. The time elapsed since the stroke ranged from 0.66 to 12.66 years (3.03 ± 2.58 years). One participant presented an outlier value (12.66 years post-stroke), which slightly increased the mean value. Among these participants, 88.46% had ischemic stroke and 11.54% had hemorrhagic stroke. Right-sided hemiparesis was more common (61.50%). According to the Modified Rankin Scale for Neurologic Disability, 23.08% of patients scored grade 1, 34.60% grade 2, 30.80% grade 3, and 11.54% grade 4, indicating that most participants presented mild to moderate disability. Most patients required a mobility aid, although nearly one third walked without assistance (30.80%). Ankle spasticity was present in all cases and interfered with daily life for the majority (73.10%). Spasticity management included botulinum toxin injections (30.80%), dry needling (11.50%), and antispastic medication (26.90%). Functional limitations were frequently reported, particularly in self-care (84.60%) and mobility (61.50%). The severity of ankle spasticity, assessed with the MAS, ranged from grade 1 to 3, with a relatively even distribution across categories. Finally, most patients had engaged in regular physical activity prior to stroke (73.10%).

### 3.2. Reliability of Ultrasound and Functional Measurements

In individuals with stroke, the measurements performed presented a good to excellent ICC (3, 1), in the affected leg for the variables Ankle Dorsiflexor (ADF) dynamometry (ICC = 0.97), Plantarflexor dynamometry (APF) (ICC = 0.96) and in the unaffected leg for the variables Dorsiflexor dynamometry (ICC = 0.94), Plantarflexor dynamometry (ICC = 0.92). In healthy subjects, a good to excellent ICC was observed in the dominant leg in the variables Dorsiflexor dynamometry (ICC = 0.91), Plantarflexor dynamometry (ICC = 0.95), Muscle thickness (ICC = 0.89), Echogenicity (ICC = 0.78), Echovariance (ICC = 0.85), Entropy (ICC = 0.85), Median (ICC = 0.76), Uniformity (ICC = 0.78), Homogeneity (ICC = 0.75), Energy (ICC = 0.76), Maximum value probability (ICC = 0.83), Sum average (ICC = 0.79), Low gray level run emphasis (ICC = 0.87), Short run low grey emphasis (ICC = 0.83), Low intensity emphasis (ICC = 0.80), Low intensity small area emphasis (ICC = 0.80), Local binary patterns (ICC = 0.85), and in the non-dominant leg in the variables Dorsiflexor dynamometry (ICC = 0.95), Plantarflexor dynamometry (ICC = 0.98), Muscle thickness (ICC = 0.77), Echovariance (ICC = 0.77), Entropy (ICC = 0.83), Uniformity (ICC = 0.75), Energy (ICC = 0.75), Maximum value probability (ICC = 0.88), Low gray level run emphasis (ICC = 0.86), Long run emphasis (ICC = 0.78, Long run low gray level emphasis (ICC = 0.79), Low intensity small area emphasis (ICC = 0.77) (see Table A1).

### 3.3. Differences in Ankle Range of Motion and Muscle Strength in Healthy and Stroke Participants

Statistically significant differences were identified in dorsiflexor and plantarflexor dynamometry, as well as dorsiflexion goniometry, between the affected limb of participants with stroke and all other limbs analyzed (unaffected limb of participants with stroke, and dominant and non-dominant limbs of healthy controls). In all cases, the affected limb showed lower values, indicating reduced strength and range of motion (Table 3). None of the covariates included in the model showed a correlation coefficient greater than 0.7, suggesting no multicollinearity issues (see Table A2).

### 3.4. Differences in Functional Mobility and Walking Speed Between Healthy and Stroke Participants

Although participants with stroke exhibited slower mean TUG and 10MWT times compared with healthy controls, these differences did not reach statistical significance. The only significant between-group difference was the higher 10MWT assistance level in stroke group (*p* = 0.002) (Table 4), indicating greater need for support during gait.

### 3.5. Differences in Ultrasound Muscle Parameters Between Healthy and Stroke Participants

No significant differences were found in structural or textural ultrasound parameters of the medial gastrocnemius between the affected and unaffected limb in stroke patients, nor between the affected limb and the dominant or non-dominant limbs of healthy controls (see Table A3).

### 3.6. Correlations Between Ultrasound Parameters and Functional Mobility, Walking Speed, Ankle Strength, Range of Motion, and Spasticity in Stroke Participants and Healthy Controls

Correlations were identified between ultrasound parameters and functional mobility, walking speed, ankle strength, range of motion, and spasticity in both stroke patients and healthy controls. Statistically significant associations are highlighted in bold in Table 5 and illustrated in Figure 1.

In the affected limb of participants with stroke, statistically significant negative correlations were observed between muscle thickness and the 10MWT at high speed (ρ = –0.266, *p* = 0.035), as well as between muscle thickness and ankle dorsiflexion goniometry (ρ = –0.025, *p* = 0.043). Additionally, echogenicity was positively correlated with ankle plantarflexion goniometry (ρ = 0.090, *p* = 0.043) and with the TUG (ρ = 0.108, *p* = 0.027), although all correlation magnitudes were negligible.

In the unaffected limb, statistically significant positive correlations were found between echogenicity and the 10MWT at high speed (ρ = 0.122, *p* = 0.022), echogenicity and TUG (ρ = 0.234, *p* = 0.046), and between echovariance and ankle plantarflexion goniometry (ρ = 0.196, *p* = 0.012), also within the negligible range.

Among healthy controls, the dominant limb showed statistically significant negative correlations between echovariance and ankle dorsiflexor goniometry (ρ = –0.097, *p* = 0.002), ankle plantarflexion goniometry (ρ = –0.467, *p* < 0.001), dorsiflexion dynamometry (ρ = –0.188, *p* = 0.003), and plantarflexor dynamometry (ρ = –0.100, *p* = 0.001). Conversely, echogenicity exhibited positive correlations with ankle dorsiflexion goniometry (ρ = 0.118, *p* = 0.027), plantarflexion goniometry (ρ = 0.453, *p* = 0.001), dorsiflexor dynamometry (ρ = 0.202, *p* = 0.005), and plantarflexor dynamometry (ρ = 0.054, *p* = 0.002), with negligible to moderate strength.

In the non-dominant limb, statistically significant negative correlations were found between echogenicity and ankle dorsiflexion goniometry (ρ = –0.188, *p* = 0.030), echovariance and ankle plantarflexion goniometry (ρ = –0.309, *p* = 0.010), and pennation angle and ankle plantarflexion goniometry (ρ = –0.109, *p* = 0.027). Statistically significant positive correlations included echogenicity with ankle plantarflexion goniometry (ρ = –0.109, *p* = 0.027) and dorsiflexor dynamometry (ρ = –0.109, *p* = 0.027), echovariance with dorsiflexion goniometry (ρ = 0.252, *p* = 0.048), muscle thickness with ankle plantarflexion goniometry (ρ = 0.030, *p* = 0.014), and pennation angle with plantarflexor dynamometry (ρ = 0.317, *p* = 0.044). All correlations ranged from negligible to moderate (Table 5, Figure 1).

### 3.7. Correlations Between Ultrasound Parameters and Functional Differences According to Spasticity Severity in Participants with Stroke

No statistically significant differences were found in ultrasound parameters (structural or textural), ankle range of motion, or muscle strength in the affected leg of participants with stroke when grouped by spasticity severity as measured by the MAS (see Table A4).

## 4. Discussion

Ultrasound and functional assessments showed high intra-rater reliability, confirming the consistency of the measurement protocol. Participants with stroke presented lower physical performance and greater functional dependence than healthy controls, reflecting the known biopsychosocial vulnerability of this population [48]. These findings provide a robust basis for interpreting subsequent comparisons between muscle characteristics and functional outcomes.

No significant differences were found in GM ultrasound parameters between the affected and unaffected limbs in participants with stroke, nor between the affected limb and the limbs of healthy controls. These findings are partially aligned with previous research describing heterogeneous echotexture patterns in neuromuscular conditions [49], but contrast with Asadi et al. [21], who identified echovariance and echogenicity as biomarkers of gastrocnemius quality after stroke. Variations in the ROI selection, image-processing methodology, or sample characteristics may partly explain these discrepancies [23]. Emerging computational approaches, including the use of machine learning for texture analysis [50], could reduce subjectivity and improve the detection of subtle structural and textural alterations that remain unnoticed with conventional techniques, thus refining the characterization of post-stroke muscle quality.

These results suggest that, in ambulatory individuals with chronic stroke, muscle structure and texture remain relatively preserved despite measurable functional deficits. This supports the hypothesis that such deficits may primarily stem from neural impairments and compensatory motor strategies rather than from overt muscle degeneration detectable by ultrasound. By contrast, significant impairments were identified in ankle dorsiflexor mobility and strength, as well as plantarflexor strength, in the affected limb compared with all other limbs. These findings, together with the preserved echotextural features, reinforce the notion that muscle performance after stroke is predominantly determined by neural mechanisms and motor unit recruitment rather than by structural muscle degeneration. Previous research also supports the limited contribution of morphology to strength generation, with neural drive and recruitment patterns being stronger determinants of functional performance [27,28].

Regarding global function, although mean TUG and 10MWT values trended toward worse performance in stroke participants (Table 4), these differences did not reach statistical significance, likely due to large interindividual variability and sample size likely limited statistical significance. Clinically, this reinforces that meaningful impairments may exist even in the absence of statistically significant group differences [51,52,53].

The correlations between GM ultrasound parameters and measures of mobility and strength were statistically significant in a few cases. Still, they remained small in magnitude, indicating that morphological features explain only a limited proportion of functional variability. Similar studies have reported inconsistent relationships between muscle echotexture and strength across different populations and muscle groups [25,54], likely due to differences in stroke phase, functional level, and the specific muscles evaluated. From a clinical perspective, although these weak associations have limited diagnostic value, they may still reflect subtle tissue adaptations. When integrated with functional assessments, echotexture analysis could contribute to the early detection of muscle quality decline in patients at risk of mobility deterioration. Ultrasound analysis complements functional tests by quantifying both structural and textural ultrasound parameters, information not captured by clinical scales or functional measures. While functional tests represent overall motor capacity, ultrasound metrics may detect early or subclinical tissue alterations, providing potential biomarkers for rehabilitation monitoring.

Regarding global function, our findings are consistent with González-Buonomo et al. (2023) [25], who also reported no significant associations between echogenicity and functional performance in ambulatory individuals with chronic stroke, but differ from Yakut et al. (2024) [26], who observed moderate correlations in acute and subacute patients. These discrepancies likely relate to differences in stroke phase, functional status, and outcome measures (TUG, 10MWT, 6MWT), as well as participants’ age distribution, factors known to influence both muscle characteristics and functional performance. It is also worth noting that the mean age of participants in our study (56.65 years) was comparable to that of González-Buonomo et al. (2023) [25] (57.4 years), whereas Yakut et al. (2024) [26] included an older sample (mean age 68.59 years). This age difference could partly explain the divergent findings, as younger individuals typically present better functional recovery, higher physical activity levels, and fewer comorbidities that may influence both muscle properties and functional outcomes.

Finally, no significant associations were found between ultrasound parameters and spasticity severity, nor did spasticity severity influence strength or range of motion in the affected limb. This reinforces the notion that post-stroke muscle alterations arise from a multifactorial interplay of disuse, compensatory motor strategies, and adaptive plasticity, rather than from spasticity alone [25,54]. This study has some limitations, including the restriction to ambulatory individuals with chronic stroke, the absence of matching for physical activity, and the lack of validated measures for diet and sleep. The heterogeneity in previous antispastic treatments (e.g., botulinum toxin A, dry needling, medication) could have influenced the assessed variables. However, this does not affect the validity of the present cross-sectional results, although such factors should be carefully controlled in future longitudinal studies aimed at analyzing treatment-related muscle changes. Moreover, the ultrasound probe was handheld rather than stabilized with a mechanical arm, which could have introduced minor variability in image acquisition. Finally, only intra-rater reliability was assessed, while inter-rater reliability was not examined, potentially limiting the generalizability of the findings to other evaluators with different training or experience levels. Strengths include recruitment from two healthcare settings, bilateral comparison with healthy controls, and all assessments performed by a single physiotherapist, enhancing reliability.

Future research should include diverse stroke phases and integrate machine learning to refine analysis, as well as exploring new assessment methodologies of ultrasound outcomes and extending the evaluation to other muscles involved in lower-limb spasticity.

Clinically, these findings emphasize that measurable strength and mobility deficits may exist even when structural and textural muscle parameters assessed by ultrasound appear preserved, underscoring the need to integrate both ultrasound and functional evaluations in post-stroke rehabilitation.

## 5. Conclusions

Structural and textural ultrasound parameters of the medial gastrocnemius did not differ between affected and unaffected limbs of stroke participants compared with healthy controls. Functional deficits were evident in ankle strength and range of motion, likely reflecting neural rather than structural mechanisms. Correlations with functional outcomes (mobility, walking speed, ankle strength, and range of motion) were small, and spasticity severity was not predictive of muscle quality. These findings highlight the need for complementary and advanced approaches to better characterize post-stroke muscle alterations and guide rehabilitation.

## Figures and Tables

**Figure 1 jcm-14-07680-f001:**
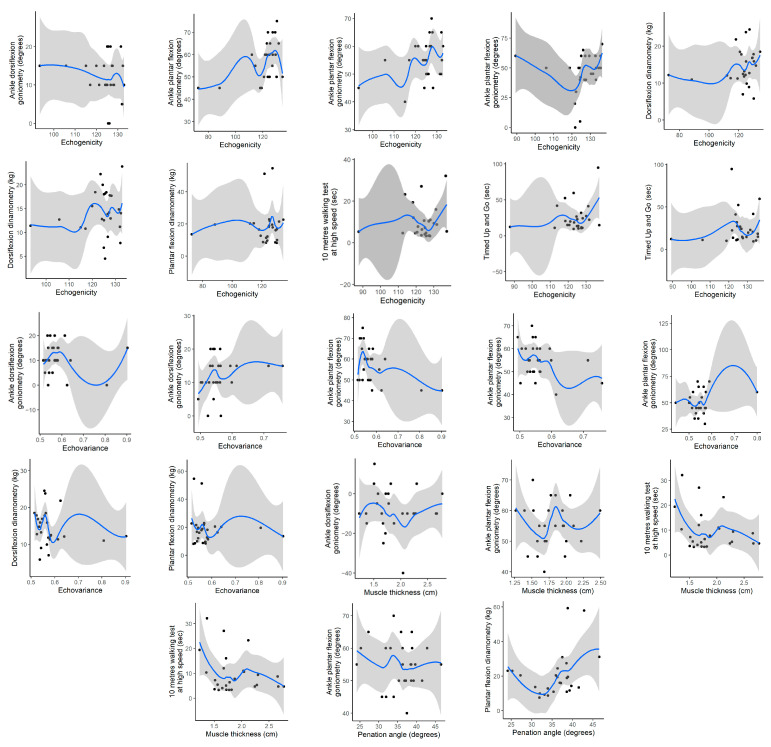
Correlations Between Ultrasound Parameters and Functional mobility, Walking speed, Ankle Strength, Range of Motion and Spasticity in Stroke Participants and Healthy Controls. Black dots represent individual observed data points. The blue line represents the smoothed trend, and the grey shaded area indicates the 95% confidence interval.

**Table 1 jcm-14-07680-t001:** Sociodemographic Anthropometric and Clinical Characteristics.

		Healthy Group	Stroke Group	^a^ *p* Value
n		26	26	
**Anthropometric and Sociodemographic characteristics**				
Age		55.96 ± 11.99	56.65 ± 11.83	0.776
Weight (kg)		73.37 ± 14.58	78.27 ± 15.97	0.355
Height (cm)		168.69 ± 9.53	169.15 ± 10.46	0.79
Dominant side, n(%)	Left	2 (7.70)	2 (7.70)	>0.999
	Right	24 (92.30)	24 (92.30)	
Biological sex, n(%)	Female	8 (30.80)	8 (30.80)	>0.999
	Male	18 (69.20)	18 (69.20)	
Educational level, n(%)	Non university	6 (23.01)	19 (73.10)	**0.001**
	University	20 (76.90)	7 (26.90)	
Marital status, n(%)	Married	10 (38.50)	16 (61.50)	0.291
	Single/divorced	13 (50.00)	9 (34.60)	
	Widow	3 (11.50)	1 (3.80)	
Live alone, n(%)	No	18 (69.20)	21 (80.80)	0.523
	Yes	8 (30.80)	5 (19.20)	
**Clinical characteristics**				
Tobacco, n(%)	Ex-smoker	7 (26.90)	11 (42.30)	0.28
	No	18 (69.20)	12 (46.20)	
	Yes	1 (3.80)	3 (11.50)	
Number of cigarettes/day		13.00 ± 7.28	21.14 ± 13.70	0.263
Alcohol, n(%)	Consumer	19 (73.10)	8 (30.80)	**0.005**
	Non consumer	7 (26.90)	18 (69.20)	
Current exercise, n(%)	No	3 (11.50)	4 (15.40)	>0.999
	Yes	23 (88.50)	22 (84.60)	
Nutritionist treatment, n(%)	No	21 (80.80)	24 (92.30)	0.419
	Yes	5 (19.20)	2 (7.70)	
Physiotherapy treatment, n(%)	No	6 (23.10)	2 (7.70)	0.248
	Yes	20 (76.90)	24 (92.30)	
Psychologist treatment, n(%)	No	11 (42.30)	7 (26.90)	0.382
	Yes	15 (57.70)	19 (73.10)	
Anxiety/depression, n(%)	No	5 (19.20)	4 (15.40)	>0.999
	Yes	21 (80.80)	22 (84.60)	
VAS-10 anxiety		3.69 ± 2.60	3.75 ± 3.06	0.919
VAS-10 depression		1.19 ± 1.74	4.08 ± 3.05	**0.001**
Physical Activity (IPAQ), n(%)	High	10 (38.50)	2 (7.70)	**0.008**
	Low	7 (26.90)	17 (65.40)	
	Moderate	9 (34.60)	7 (26.90)	

Data expressed with mean ± standard deviation or with relative absolute values (%); VAS: Visual Analogue Scale; IPAQ: International Physical Activity Questionnaire. ^a^ significant if *p* < 0.05 (shown in bold).

**Table 2 jcm-14-07680-t002:** Description of stroke-specific characteristics.

Stroke-Specific Characteristics		
Time since stroke, years (mean ± SD)	3.03 ± 2.58	
Type of stroke	Ischemic	23 (88.46)
	Hemorrhagic	3 (11.54)
Hemiparetic side, n(%)	Left	10 (38.50)
	Right	16 (61.50)
Modified Ranking Scale for Neurologic Disability	0	0 (0.00)
	1	6 (23.08)
	2	9 (34.60)
	3	8 (30.80)
	4	3 (11.54)
	5	0 (0.00)
	6	0 (0.00)
Walking assistive device, n(%)	Cane	11 (42.30)
	Cane and Rancho Los Amigos splint	5 (19.20)
	Wheelchair	2 (7.70)
	No	8 (30.80)
Patients’ perception of daily life ankle spasticity interference, n(%)	No	7 (26.90)
	Yes	19 (73.10)
Botulinum toxin, n(%)	No	18 (69.20)
	Yes	8 (30.80)
Dry needling, n(%)	No	23 (88.50)
	Yes	3 (11.50)
Spasticity medication, n(%)	No	19 (73.10)
	Yes	7 (26.90)
Mobility difficulties, n(%)	No	10 (38.50)
	Yes	16 (61.50)
Self-care difficulties, n(%)	No	4 (15.40)
	Yes	22 (84.60)
Leisure/hobbies/work difficulties, n(%)	No	8 (30.80)
	Yes	18 (69.20)
Modified Ashworth Scale, n(%)	1	8 (30.80)
	1+	6 (23.10)
	2	6 (23.10)
	3	6 (23.10)
	None	0 (0.00)
Exercise before stroke, n(%)	No	7 (26.90)
	Yes	19 (73.10)

**Table 3 jcm-14-07680-t003:** Ankle Range of Motion and Strength Parameters in Stroke and Healthy Participants.

	Stroke Affected Leg	Stroke Non Affected Leg	Healthy Dominant Leg	Healthy Non Dominant Leg	Stroke affected vs. non affected leg (^a^ *p* value)	Stroke Affected Leg vs. Healthy Dominant Leg (^a^ *p* Value)	Stroke Affected Leg vs. Healthy Non Dominant Leg (^a^ *p* Value)
ADF goniometry (°)	−8.46 ± 11.11	5.96 ± 9.49	11.15 ± 6.05	11.92 ± 5.49	15.26 (SE = 3.23), t = 4.72, *p* **< 0.001**	18.53 (SE = 3.97), t = 4.66, *p* **< 0.001**	19.47 (SE = 3.77), t = 5.16, *p* **< 0.001**
APF goniometry (°)	45.85 ± 16.87	50.19 ± 10.63	57.31 ± 8.86	55.00 ± 7.35	7.53 (SE = 5.43), t = 1.39, *p* = 0.18	13.20 (SE = 7.68), t = 1.72, *p* = 0.09	9.47 (SE = 7.34), t = 1.29, *p* = 0.21
ADF dynamometry (kg)	3.27 ± 4.06	9.54 ± 4.64	14.43 ± 4.73	14.25 ± 4.68	5.81 (SE = 1.49), t = 3.9, *p* = **0.001**	11.58 (SE = 2.45), t = 4.72, *p* **< 0.001**	11.34 (SE = 2.43), t = 4.65, *p* **< 0.001**
APF dynamometry (kg)	5.80 ± 6.07	11.23 ± 6.56	18.56 ± 11.23	20.46 ± 13.02	4.95 (SE = 2.01), t = 2.47, *p* = **0.02**	9.10 (SE = 4.81), t = 1.89, *p* = 0.07	12.48 (SE = 4.51), t = 2.77, *p* = **0.009**

ADF: Ankle Dorsiflexor; APF: Ankle Plantarflexor; SE: Standard error; Data expressed with mean ± standard deviation. ^a^ significant if *p* < 0.05 (shown in bold).

**Table 4 jcm-14-07680-t004:** Functional Mobility and Walking Speed Outcomes in Stroke and Healthy Participants.

Functional Mobility and Walking Speed	Healthy Group	Stroke Group	^a^ *p* Value
TUG (s)	8.29 ± 0.96	25.04 ± 19.47	9.84 (SE = 7.82), t = 1.26, *p* = 0.227
10MWT at normal speed (s)	4.60 ± 0.65	11.44 ± 8.33	4.21 (SE = 3.24), t = 1.30, *p* = 0.202
10MWT at high speed (s)	3.11 ± 0.43	9.71 ± 7.79	3.56 (SE = 3.12), t = 1.14, *p* = 0.263
10MWT assistance level	7.00 ± 0.00	6.15 ± 0.88	−1.04 (SE = 0,31), t = −3.32, *p* = **0.002**

Data expressed with mean ± standard deviation or with relative absolute values (%). ^a^ significant if *p* < 0.05 (shown in bold).

**Table 5 jcm-14-07680-t005:** Correlations Between Ultrasound Parameters and Functional mobility, Walking speed, Ankle Strength, Range of Motion and Spasticity in Stroke Participants and Healthy Controls.

		Stroke Affected Leg ρ (^a^ *p* Value)	Stroke Non Affected Leg ρ (^a^ *p* Value)	Healthy Dominant Leg ρ (^a^ *p* Value)	Healthy Non Dominant Leg ρ (^a^ *p* Value)
**Echogenicity**	Ankle dorsiflexion goniometry (°)	0.087, *p* = 0.171	0.083, *p* = 0.258	0.118, *p* = **0.027**	−0.188, *p* = **0.03**
	Ankle plantarflexion goniometry (°)	0.09, *p* = **0.043**	−0.208, *p* = 0.194	0.453, *p* = **0.001**	0.339, *p* = **0.001**
	Dorsiflexor dynamometry (kg)	0.143, *p* = 0.223	−0.025, *p* = 0.185	0.202, *p* = **0.005**	0.162, *p* = **0.009**
	Plantarflexor dynamometry (kg)	− 0.002, *p* = 0.148	−0.094, *p* = 0.153	0.054, *p* = **0.002**	−0.117, *p* = 0.065
	10MWT at normal speed (s)	0.207, *p* = 0.07	0.124, *p* = 0.11		
	10MWT at high speed (s)	0.153, *p* = 0.34	0.122, *p* = **0.022**		
	TUG (s)	0.108, *p* = **0.027**	0.234, *p* = **0.046**		
	MAS	0.041, *p* = 0.17	0.116, *p* = 0.185		
**Echovariance**	Ankle dorsiflexion goniometry (°)	0.164, *p* = 0.099	−0.149, *p* = 0.341	−0.097, *p* = **0.002**	0.252, *p* = **0.048**
	Ankle plantarflexion goniometry (°)	−0.032, *p* = 0.461	0.196, *p* = **0.012**	−0.467, *p* **< 0.001**	−0.309, *p* = **0.01**
	Dorsiflexor dynamometry (kg)	0.061, *p* = 0.114	0.004, *p* = 0.062	−0.188, *p* = **0.003**	−0.181, *p* = 0.08
	Plantarflexor dynamometry (kg)	0.169, *p* = 0.493	0.027, *p* = 0.225	−0.1, *p* = **0.001**	−0.012, *p* = 0.145
	10MWT at normal speed (s)	−0.322, *p* = 0.161	−0.248, *p* = 0.151		
	10MWT at high speed (s)	−0.276, *p* = 0.162	−0.257, *p* = 0.229		
	TUG (s)	−0.138, *p* = 0.082	−0.218, *p* = 0.072		
	MAS	−0.152, *p* = 0.396	−0.106, *p* = 0.086		
**Muscle** **thickness (cm)**	Ankle dorsiflexion goniometry (°)	−0.025, *p* = **0.043**	0.06, *p* = 0.742	0.016, *p* = 0.704	−0.139, *p* = 0.39
	Ankle plantarflexion goniometry (°)	−0.371, *p* = 0.224	0.207, *p* = 0.45	−0.077, *p* = 0.366	0.03, *p* = **0.014**
	Dorsiflexor dynamometry (kg)	−0.239, *p* = 0.491	0.315, *p* = 0.264	0.116, *p* = 0.284	−0.013, *p* = 0.534
	Plantarflexor dynamometry (kg)	−0.125, *p* = 0.519	0.142, *p* = 0.942	0.532, *p* = 0.236	0.412, *p* = 0.28
	10MWT at normal speed (sec)	−0.279, *p* = 0.669	0.102, *p* = 0.607		
	10MWT at high speed (s)	−0.266, *p* = **0.035**	0.096, *p* = 0.325		
	TUG (s)	−0.332, *p* = 0.182	−0.062, *p* = 0.367		
	Modified Ashworth Scale (MAS)	−0.038, *p* = 0.317	−0.041, *p* = 0.383		
**Penation angle** (°)	Ankle dorsiflexion goniometry (°)	0.306, *p* = 0.143	0.074, *p* = 0.484	−0.079, *p* = 0.075	−0.039, *p* = 0.096
	Ankle plantarflexion goniometry (°)	−0.344, *p* = 0.252	0.14, *p* = 0.088	−0.295, *p* = 0.486	−0.109, *p* = **0.027**
	Dorsiflexor dynamometry (kg)	0.069, *p* = 0.179	0.206, *p* = 0.232	−0.198, *p* = 0.285	−0.274, *p* = 0.183
	Plantarflexor dynamometry (kg)	0.358, *p* = 0.544	0.54, *p* = 0.15	0.171, *p* = 0.541	0.317, *p* = **0.044**
	10MWT at normal speed (s)	−0.083, *p* = 0.474	−0.24, *p* = 0.485		
	10MWT at high speed (s)	−0.048, *p* = 0.543	−0.247, *p* = 0.09		
	TUG (s)	0.037, *p* = 0.075	−0.243, *p* = 0.121		
	MAS	−0.439, *p* = 0.719	−0.399, *p* = 0.051		

^a^ significant if *p* < 0.05 (shown in bold).

## Data Availability

The data that support the findings of this study are available from the corresponding author upon reasonable request. Due to ethical and privacy restrictions, the raw data are not publicly available.

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
