# Peer review of "Structural and Textural Ultrasound Features of Gastrocnemius Medialis in Chronic Stroke: Associations with Functional Outcomes and Spasticity"

_jcm, 2025, doi:10.3390/jcm14217680_

Round 1

Reviewer 1 Report

Comments and Suggestions for Authors

Dear Authors,

I appreciate the opportunity to review this report on comparison of structural and textual ultrasound features of the gastrocnemius medialis between chronic stroke patients and healthy controls.

Good description of analysis plan, as well as methodology.

My comments:

In your work you analyze chronic stroke patients, and in M&M you state that one of inclusion criteria was: “at least 6 months post-stroke”.

As chronic stage of stroke begins 6 months after the initial event, it would be interesting to state the time range after the stroke in study/stroke group.

Among those 26 patients in study/stroke group, it would be interesting to provide the percentage of patients with ischemic and hemorrhagic stroke.

Regarding lifestyle habits, patients with stroke reported lower alcohol consumption? When? before or after their stroke, please clarify.

M&M

Line 121 IRENEA would be sufficient

Reviewer 2 Report

Comments and Suggestions for Authors

This an interesting manuscript however several doubts must be resolved.

According to the Declaration of Helsinki. What year it was approved?

Regarding to the inclusion criteria of the stroke group, Was it taken into account whether patients had received botulinum toxin or dry needling at least three months prior to entering the study? Since this could alter the results. The Modified Ashworth Scale, a reference is needed the first time that it is mentioned.  In addition the Timed Up and Go (TUG) and the 10-Meter Walk Test (10MWT); ankle muscle strength during maximal isometric contractions of dorsiflexion need references And how were they carried out? For example, how was the 10MWT carried out? Was a distance added for acceleration and another for deceleration and plantarflexion, assessed with a MicroFET2 dynamometer? How were the MAS, goniometry, and other scales and tests performed? In general, this section needs to provide more in-depth details on all the procedures for assessing the different variables.
Correlation strength was classified as negligible (<0.29), low (0.30–0.49), moderate (0.50–0.69), high (0.70–0.89), or very high (>0.90) this needs a reference.

This section need improvement "Data Protection Data handling adhered to General Data Protection Regulation (2016/679) and Spanish Organic Law 3/2018. Pseudonymization was performed by a non-study iHealthy team member using randomly assigned numeric codes".

Regarding Table 1, I do not see the gender of the patients and the control group. Were they balanced? The degree of impairment in individuals with stroke is not described as well as the time since stroke. Was it the first stroke or had they suffered more? What was the degree of impairment before the stroke? How were they according to the mRankin scale?

Regarding Table 2, do the MAS data correspond to the ankle? Why did the authors not evaluate the knee, given that the gastrocnemius muscle inserts into the femoral condyle?

From my point of view, the stroke patients were young, considering that the average age in Spain is 70-72 years old. This could influence the results.

Table 4 The 10MWT is usually expressed in m/s. What do the authors mean as 10MWT assistance level?

Discussion

No changes are found in the GM, but the problem is that we do not know how much time has passed since the patients suffered a stroke. They are in the subacute or chronic phase, as this can make a big difference.  We also do not know how many women there were.

Round 2

Reviewer 2 Report

Comments and Suggestions for Authors

the authors have made the suggested changes